# Immune-related pan-cancer gene expression signatures of patient survival revealed by NanoString-based analyses

Alberto D'Angelo[1,2☯*], Huseyin Kilili[3☯], Robert Chapman[4], Daniele Generali[5], Ingeborg Tinhofer[6], Stefano Luminari[7,8], Benedetta Donati[9], Alessia Ciarrocchi[9], Riccardo Giannini[10], Roberto Moretto[11], Chiara Cremolini[12], Filippo Pietrantonio[13], Navid Sobhani[14], Debora Bonazza[15], Robert Prins[16], Seung Geun Song[17], Yoon Kyung Jeon[17,18], Giuseppina Pisignano[1], Mattia Cinelli[1], Stefan Bagby[1], Araxi O. Urrutia[3,19]

1 Department of Life Sciences, University of Bath, Bath, United Kingdom, 2 Oncology Department, Royal United Hospital, Bath, United Kingdom, 3 Milner Centre, Department of Life Sciences, University of Bath, Bath, United Kingdom, 4 Department of Medicine, The Princess Alexandra Hospital, Harlow, United Kingdom, 5 Multidisciplinary Unit of Breast Pathology and Translational Research, Cremona Hospital, Cremona, Italy, 6 Department of Radiooncology and Radiotherapy, Charite´ University Hospital, Berlin, Germany, 7 Hematology Unit, Azienda USL-IRCCS, Reggio Emilia, Italy, 8 Surgical, Medical and Dental Department of Morphological Sciences Related to Transplant, Oncology and Regenerative Medicine, University of Modena and Reggio Emilia, Reggio Emilia, Italy, 9 Translational Research Laboratory, Azienda USL-IRCCS, Reggio Emilia, Italy, 10 Department of Surgery, Clinical, Molecular and Critical Care Pathology, University of Pisa, Pisa, Italy, 11 Unit of Medical Oncology 2, Azienda Ospedaliero-Universitaria Pisana, Pisa, Italy, 12 Department of Translational Research and New Technologies in Medicine and Surgery, University of Pisa, Pisa, Italy, 13 Department of Oncology and Heamto-Oncology, University of Milan, Milan, Italy, 14 Section of Epidemiology and Population Science, Department of Medicine, Baylor College of Medicine, Houston, Texas, United States of America, 15 Department of Medical, Surgical and Health Sciences, Cattinara Hospital, University of Trieste, Trieste, Italy, 16 Department of Neurosurgery, David Geffen School of Medicine, University of California, Los Angeles, Los Angeles, California, United States of America, 17 Department of Pathology, Seoul National University College of Medicine, Seoul, Republic of Korea, 18 Cancer Research Institute, Seoul National University, Seoul, Republic of Korea, 19 Instituto de Ecologia, UNAM, Ciudad de Mexico, Mexico

☯ These authors contributed equally to this work.
* ada43@bath.ac.uk

**Data Availability Statement:** Dataset GSE numbers included in this study are reported in supplementary Table 2. Datasets for melanoma (row 2), ovarian cancer (row 3), head and neck

## Abstract

The immune system plays a central role in the onset and progression of cancer. A better understanding of transcriptional changes in immune cell-related genes associated with cancer progression, and their significance in disease prognosis, is therefore needed. Nano-String-based targeted gene expression profiling has advantages for deployment in a clinical setting over RNA-seq technologies. We analysed NanoString PanCancer Immune Profiling panel gene expression data encompassing 770 genes, and overall survival data, from multiple previous studies covering 10 different cancer types, including solid and blood malignancies, across 515 patients. This analysis revealed an immune gene signature comprising 39 genes that were upregulated in those patients with shorter overall survival; of these 39 genes, three (MAGEC2, SSX1 and ULBP2) were common to both solid and blood malignancies. Most of the genes identified have previously been reported as relevant in one or more cancer types. Using Cibersort, we investigated immune cell levels within individual cancer types and across groups of cancers, as well as in shorter and longer overall survival groups.

cancer (1st dataset, row 4), pancreatic cancer (row 5), lung cancer (row 8) and large B cell lymphoma (row 9) are publicly available and can be downloaded at the following GSE numbers, respectively: GSE124574, EGAS00001002839, GSE122272, GSE132946, GSE161116, GSE102818, GSE147115/GSE147116. The authors do not have the right to share gene expression datasets of glioblastoma (row 1), head and neck (2nd dataset, row 4), colon cancer (row 7) and Hodgkin lymphoma (row 10). To access these publicly unavailable datasets, please contact: braintumor@mednet.ucla.edu (glioblastoma dataset - row 1), technologietransfer@Dkfz-Heidelberg.de (head and neck cancer, 2nd dataset - row 4), international@med.unipi.it (colon cancer dataset - row 7) and urp@unimore.it (Hodgkin lymphoma dataset - row 10).

**Funding:** The authors received no specific funding for this work.

**Competing interests:** The authors have declared that no competing interests exist.

Patients with shorter survival had a higher proportion of M2 macrophages and γδ T cells. Patients with longer overall survival had a higher proportion of CD8+ T cells, CD4+ T memory cells, NK cells and, unexpectedly, T regulatory cells. Using a transcriptomics platform with certain advantages for deployment in a clinical setting, our multi-cancer meta-analysis of immune gene expression and overall survival data has identified a specific transcriptional profile associated with poor overall survival.

## 1. Introduction

Cancer cells use a range of immune suppression and evasion functions to wrest control of the tumour microenvironment [1], including upregulation of genes such as immune checkpoints [2]. Immunotherapies, on the other hand, are designed to fight cancer cells by boosting particular immune system components.

Despite recent progress in cancer immunotherapies such as immune checkpoint inhibition, response rates vary between malignancies and even within patient cohorts diagnosed with the same malignancy [3–5]. Advanced/metastatic cancer patients, moreover, exhibit little response to immunotherapies whilst enduring a high risk of toxicity exposure [6]. Such variable success is probably linked to the complexity of the tumour microenvironment, which involves cell-cell interactions among multiple cell types, with accompanying dynamic genomic and epigenetic characteristics [7–9]. Systematic investigation of immune gene expression data across a broad spectrum of malignancies can improve understanding of the complex anticancer responses of the immune system, facilitating elucidation of more effective immunotherapies [10, 11]. Over-all survival is a key parameter in cancer management, for example in making treatment decisions and quality of life assessments, and appraising effectiveness of the healthcare system. Identification of transcriptomic immune signatures associated with overall survival is therefore a crucial goal in cancer genomics.

Using RNA sequencing data, several studies have identified prognostic gene expression signatures in a single malignancy [12–20] or across multiple malignancies [21–26]. These efforts have focused on examining transcriptomic signatures according to patient response to immune checkpoint inhibitors [22], investigating genes associated with the cell cycle [23], extracellular matrix [24], melatonergic [25] and WNT [26] pathways. Overall survival-associated gene expression signatures obtained using RNA-seq gene expression data have shown statistical reliability in predicting patient outcome [27, 28]. Several studies have examined immune system gene signatures associated with overall survival across multiple cancer types, including assessment of solid tumours according to response to immune checkpoint inhibitors [29], tumours displaying specific tumour cell attributes such as high levels of transforming growth factor-beta [30], the prognostic significance of previously established cancer hallmark genes [31], and machine learning approaches [32–34]. Gene expression data analysis is thus a powerful way to investigate the molecular and cellular mechanisms underlying disease progression and patient overall survival, including the role of the immune system in these outcomes [33].

Development of techniques and treatment plans that take into account patient-specific immunological profiles, however, requires that patient transcriptional profiles in clinical settings are obtained in large numbers. Widespread implementation of RNA-seq gene expression profiling in clinical settings is currently difficult, for example due to cost. NanoString is an alternative for transcriptomic investigation with some advantages over RNA-seq gene expression profiling approaches, including direct quantification of target molecules with digital

precision, incorporation of a standard, a consistent number of genes per panel (no need for further validation), and no requirement for amplification steps or replicates; these characteristics should reduce artificial bias [35, 36]. Additionally, NanoString does not require library construction or enzymatic steps (no reverse transcription) and uses standard workflows and analysis pipelines [35, 36]. Together with its capacity to exploit formalin-fixed and paraffin-embedded (FFPE) samples, these characteristics make NanoString technology potentially useful in a clinical setting [35, 36]. More specifically, the NanoString PanCancer Immune Profiling panel simultaneously analyses 770 human immune-related genes using 100-mer colour-coded barcodes that represent single target transcripts.

NanoString-based studies of various tumour features [37–41] have identified prognostic gene expression signatures in a single malignancy [42–52]. Studies focused on identification of prognostic immune signatures in more than one malignancy using NanoString technology remain scarce and, so far, efforts to characterize overall survival signatures have focused on patients affected by a specific complication or associated with a specific genomic alteration. Wu et al [53] investigated 155 cancer patients diagnosed with thoracic cancers leading to malignant pleural effusion. Reckamp et al [54] investigated 21 patients diagnosed with five different malignancies, all associated with MET gene alteration. No studies to date have investigated common immune gene signatures using the NanoString PanCancer Immune Profiling panel.

Here we perform a comprehensive NanoString-based analysis of immune gene expression and cell population profiling of 515 patients diagnosed with 10 different cancer types, including solid tumours and blood malignancies, with associated overall survival data, with the aim of identifying common prognostic immune gene signatures. We hypothesized that such an immune gene signature, associated with patient overall survival regardless of cancer type, would constitute a prognostic parameter.

We have identified a prognostic immune gene signature using the NanoString PanCancer Immune Profiling panel. We believe that combining a pan-cancer approach with expression data for 770 immune-related genes and associated overall survival data is a robust, effective strategy for the identification of prognostic immune gene signatures.

## 2. Methods

### 2.1. Expression datasets and cancer types

All datasets were obtained using the NanoString Immune PanCancer Profiling panel, which comprises 770 immune-related genes, and included associated clinical data of 515 patients encompassing 10 solid and blood cancer types: pancreatic cancer (n = 7) [42], melanoma (n = 19) [38], ovarian cancer (n = 20) [55], breast cancer (n = 32) [56], head and neck cancer (n = 80 across two cohorts) [57, 58], colon cancer (n = 89) [59], glioblastoma (n = 29) [60], lung cancer (n = 17) [61], large B cell lymphoma (n = 50) [62], and Hodgkin lymphoma (n = 172) [63] (S1 Table). Where publicly available, anonymised clinical data and gene expression datasets as raw counts (RCC files) were downloaded from GEO platforms (S2 Table). If data were not publicly available, the relevant corresponding authors were contacted and asked to share their gene expression and clinical data. All expression datasets were then combined into a single dataset. The included cancer patient characteristics are described in S1 Table. We included all patients (n = 515) in all of our analyses.

### 2.2. Overall survival for each cancer type

Violin plots of overall survival were constructed using the ggplot package [64] in R. Probability of overall survival as a function of time across cancer types was calculated using Kaplan-Meier log-ranks [65, 66].

## 2.3. Transcriptome profile similarity analyses

Multidimensional scaling (MDS) plots were used to observe the level of similarity in gene expression patterns among patients [67, 68]. Analysis of Similarity (ANOSIM) [69, 70] was used to establish whether clusters from the MDS plots were significant. The pheatmap R package [71] was used to plot a heatmap of gene expression of 770 genes across all patients, with the heatmap scaled according to gene expression. We also selected groups of genes tracking several cell types (e.g. genes tracking CD8+ T cells, CD4+ activated T cells, NK cells, B cells) and immune-related genes (TIM3, LAG3, CTLA4, PDCD1, TIGIT) that have been identified as important for their prognostic and therapeutic value in cancer [72]; given their significance, we generated a heatmap for each of these groups of genes.

## 2.4. Differential gene expression analysis

In order to identify genes with increased and decreased expression, differential gene expression analysis was carried out using the edgeR package [67, 68]. Gene expression data were normalized with the TMM method and scaled according to coding sequence length. Filtration was performed to determine which genes have sufficient counts to be passed onto statistical analysis; genes with at least 1 count-per-million (cpm) in a group were retained and lowly expressed genes were discarded as explained by Chen et al [73]. We then assigned long, medium and short overall survival rates to each patient according to their cancer type. For each cancer type, we then obtained the overall survival rate separately and the highest 20% and lowest 20% overall survival rates were classified as long and short overall survival, respectively; the remaining survival rates were classified as medium survival. As we have combined datasets from different studies, we introduced a batch effect to the edgeR model to reduce inaccurate results. The glmQLF Test was then used to detect significantly differentially expressed genes between short and long overall survival patients. We set the significance threshold to 0.5 log fold change (log FC) and false discovery rate (FDR) < 0.05 for both upregulated and downregulated genes in order to capture genes with greater fold change from the list of genes of interest. Volcano plots were generated using EnhancedVolcano [74] to visualize the differential gene expression analysis results.

## 2.5. Protein-protein interaction

Protein-protein interactions involving shorter overall survival-associated genes were identified using STRING, a biological database and visualization tool for network analyses [75]. A wider set of interactions was displayed by expanding protein-protein interaction networks three times.

## 2.6. Estimation of relative levels of immune cell types

Cibersort [76] was used to estimate immune cell levels based on patient NanoString gene expression profiles. Cibersort uses a deconvolution algorithm, built on nine normalized gene expression profiles, to characterize immune-related cell composition of tissues. We used Cibersort and leukocyte signature matrix 22 (LM22) to quantify the proportions of immune cell types from cancer patient NanoString gene expression data. Normalised gene expression data were evaluated using the Cibersort algorithm, running 1000 permutations. Samples with Cibersort p-value below the recommended threshold of 0.05 were included in correlation analyses between gene expression and immune cell types.

## 3. Results

### 3.1. Cohort description and data exploratory analysis

Gene expression and overall survival data for 515 cancer patients were used. We divided cancer types into two groups: solid malignancies (breast, colon, head and neck, lung, ovarian and pancreatic cancers, glioblastoma and melanoma) and blood malignancies (B cell lymphoma and Hodgkin lymphoma) (S1 Table), comprising 56.9% and 43.1% of the total cohort, respectively (S1 Table). Among solid malignancies, the largest and smallest groups were colon cancer and pancreatic cancer at 17.3% and 1.4% of the total cohort, respectively. The cohort of 515 patients encompassed large variations in overall survival, with glioblastoma and melanoma patients having the shortest median mean overall survival rates at 9.5 and 17.3 months, respectively. Colon and ovarian cancer patients had the longest median overall survival rates at 48.3 and 56.4 months, respectively (S1 Table; Fig 1A). According to Kaplan-Meier log-rank estimates, glioblastoma and melanoma have lower overall survival probability than the other cancer types (Fig 1B).

### 3.2. Gene expression similarity by cancer type, cancer class and overall survival

We performed MDS analysis followed by ANOSIM test to visualise similarities and dissimilarities of gene expression levels for each patient when grouped into cancer type, cancer class and survival (Fig 2). Significant clustering was observed according to cancer type (ANOSIM with 1000 permutations, P < 0.0001, R = 0.7) (Fig 2A) and cancer class (solid vs blood) (ANOSIM with 1000 permutations, P < 0.0001, R = 0.5) (Fig 2B). In contrast, significant clustering was not observed when patients were classified according to overall survival (long, medium and short) (ANOSIM P = 0.68) (Fig 2C). Significant clustering in Fig 2 (panels A and B) suggests that patients with the same cancer types and class (solid or blood) tend to have similar gene expression levels. Such similarity between gene expression levels was not observed when patients were grouped according to survival rate (Fig 2C).

### 3.3. Cancer type-specific immune gene expression profiles

We observed cancer type- and cancer class-specific gene expression profiles (Fig 3). Within this, marked differences were observed in five gene sets of relevance to immunotherapy. The CD8+ T cell set (Fig 3) comprises genes that activate CD8+ cells and granzyme–mediated apoptosis pathways (e.g. GZMB, CD8A, PRF1, CD8B). The CD4+ activated T cell set comprises IL26 and IL17A, activating genes for T helper cells. The NK cell set comprises activating genes for NK cells: NCR1, KLRB1, KLRC1 and KLRD. The B cell set comprises B cell-activating genes: BLK, CD19 and MS4A1. The immune checkpoint inhibitor set comprises checkpoint inhibitors that are targeted in current therapies: TIM-3, LAG3, CTLA4, PD1 and TIGIT.

### 3.4. Differential gene expression analysis

We performed differential gene expression analysis to identify genes associated with patient survival (Fig 4A–4C). 39 genes were found to be significantly upregulated and eight genes downregulated in short overall survival patients (S3 Table). As we observed clustering between solid and blood cancer patients in the MDS plot (Fig 2B), we performed two additional differential gene expression analyses, one with only solid cancer patients and one with only blood cancer patients. For solid cancer patients, 22 genes were upregulated and six genes were downregulated in short overall survival patients compared to long overall survival patients (Fig 4B) (S4 Table). For blood cancer patients, 55 genes were upregulated and 23 genes were

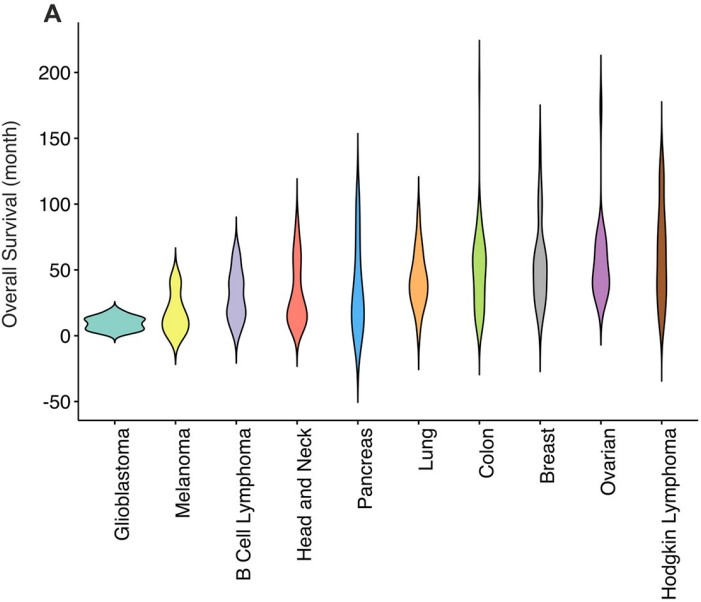

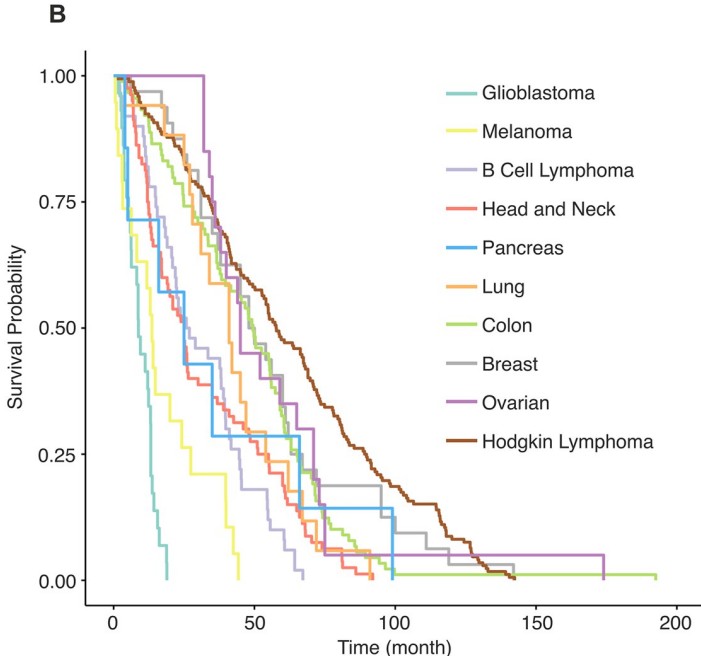

**Fig 1. Overall survival and survival probability by cancer type.** (A) A violin plot showing the distribution of overall survival (OS) rate for all 515 patients grouped according to their cancer type. OS rate of patients is shown in months on the y-axis. Cancer types are shown on the x-axis and are ordered by average overall survival rate for each cancer type, from lowest to the highest. (B) Kaplan-Meier estimation curves show survival probability of cancer types. Each curve represents a different cancer type (Kaplan-Meier log-rank test, P < 0.0001). Both panel A and panel B show that glioblastoma and melanoma have the lowest survival rates whereas patients with colon, breast and ovarian cancers, and Hodgkin lymphoma, tend to live longer.

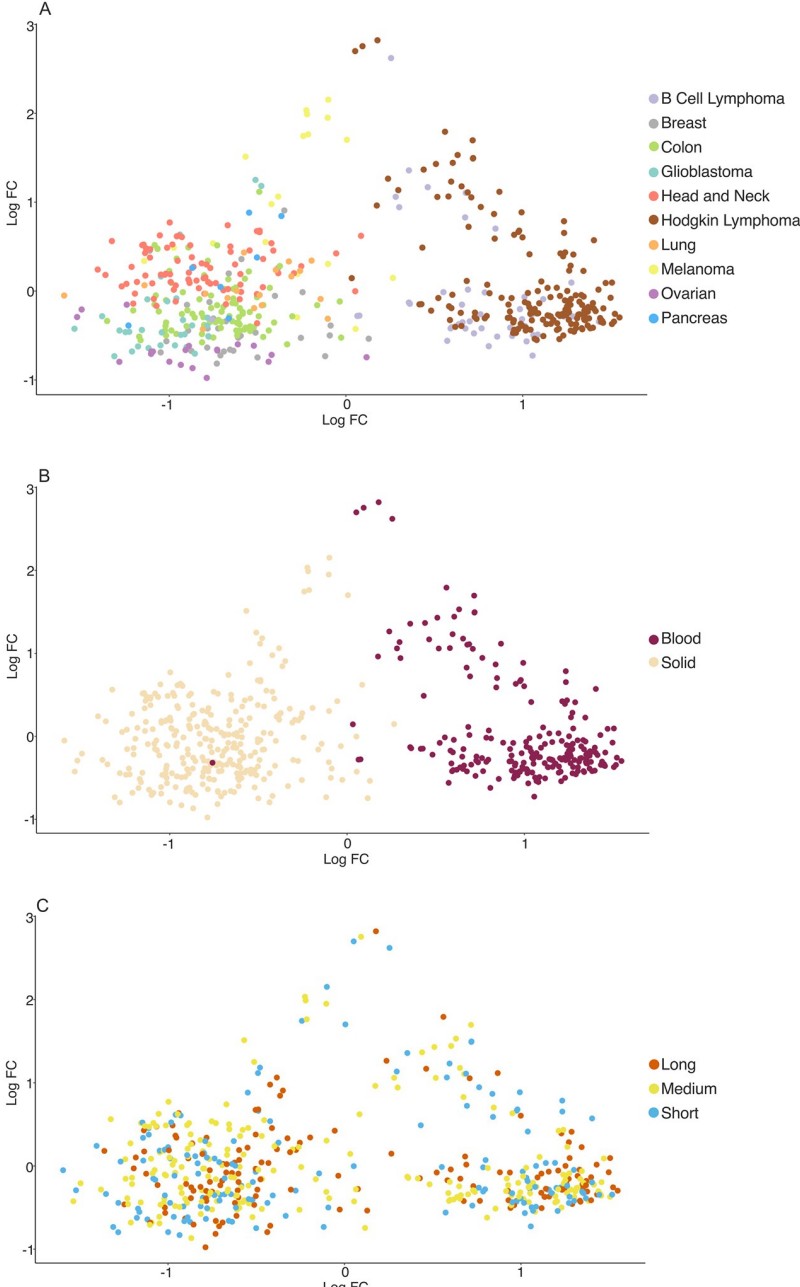

**Fig 2. Clustering by cancer type, classification and survival.** Multidimensional scaling (MDS) plots showing dissimilarity of patients based on their gene expression profiles (panels A, B and C). (A) MDS plots of patients grouped according to cancer type (ANOSIM with 1000 permutations, P < 0.0001, R = 0.7). (B) MDS plot of patients grouped according to cancer classification (solid and blood: ANOSIM with 1000 permutations, P < 0.0001, R = 0.5). (C) MDS plot of patients grouped according to survival rate (ANOSIM with 1000 permutations, P > 0.05). The clustering observed in panels A and B indicates that patients with the same cancer type and/or cancer class (solid and blood malignancies) tend to show similar gene expression levels. However, no clustering is observed in panel C, indicating that gene expression level and survival rate are not correlated.

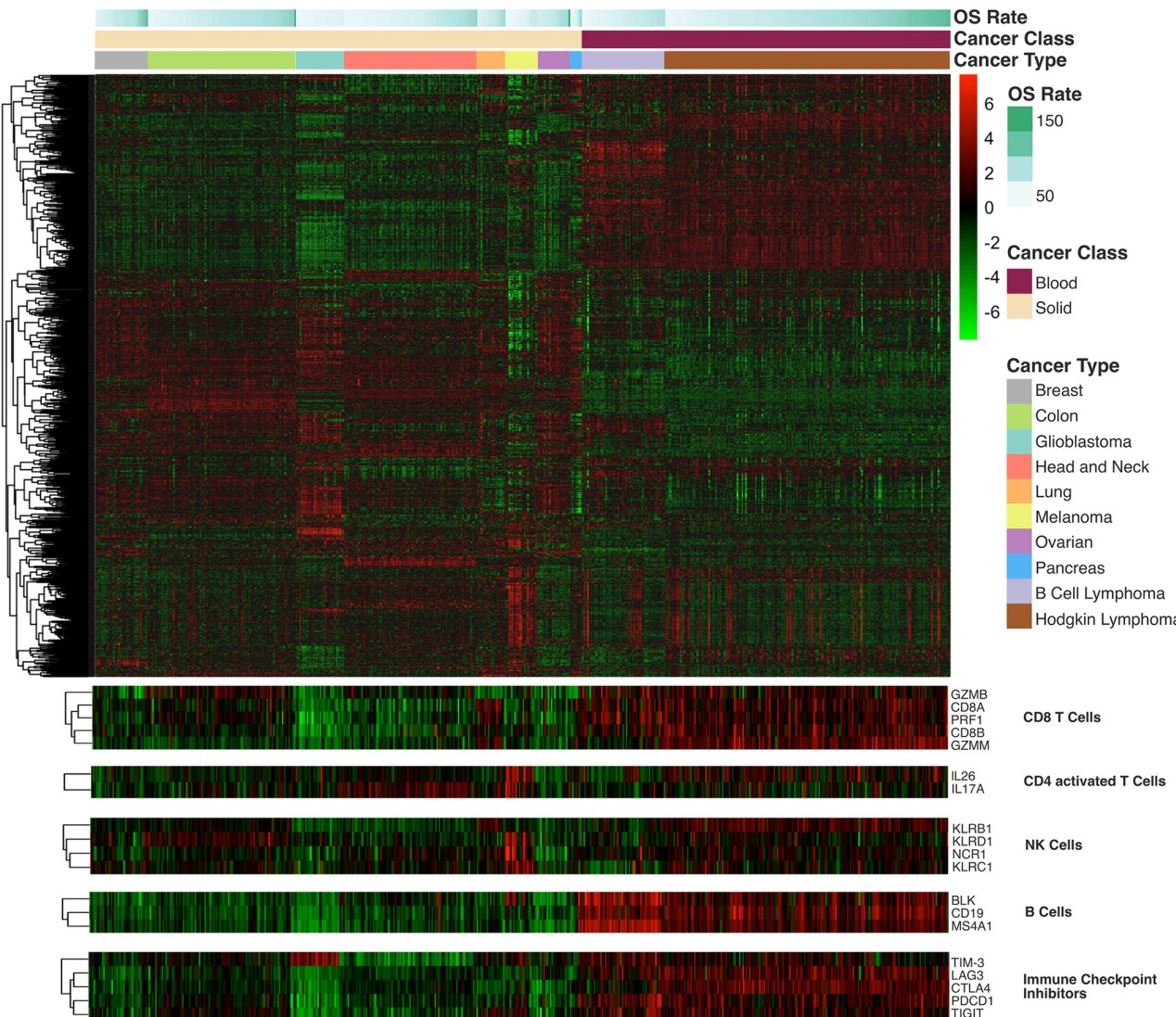

**Fig 3. Heatmap of immune-related gene expression levels across cancer types.** The heatmap shows hierarchical clustering of 770 immune-related genes in all 515 patients across the 10 different malignancies considered in this study. Green and red indicate downregulation and upregulation of gene expression, respectively, with intensity reflecting degree of change. The three bars above the heatmap indicate overall survival (OS) rate, cancer class and cancer type, with colour codes indicated on the right side of the figure. The darker the colour in the OS rate bar, the longer the OS rate. In the cancer class bar, beige represents solid malignancies and purple represents blood malignancies. Each cancer type is assigned a colour shown in the cancer type bar. Below the heatmap, five handpicked immune clusters of interest are reported; it is clear that these genes of interest are upregulated in blood malignancies.

downregulated in short overall survival patients (Fig 4C) (S5 Table). Three upregulated genes were common to both solid and blood cancers (MAGEC2, SSX1, ULBP2) (Fig 5A), but there were no common downregulated genes (Fig 5B).

## 3.5. Protein-protein interaction networks involving genes associated with cancer overall survival

Genes upregulated in short overall survival cancer patients were further characterised using the STRING protein-protein interaction database. Among the 39 genes upregulated within all cancers, two protein-protein interaction networks were observed (Fig 6): one composed primarily

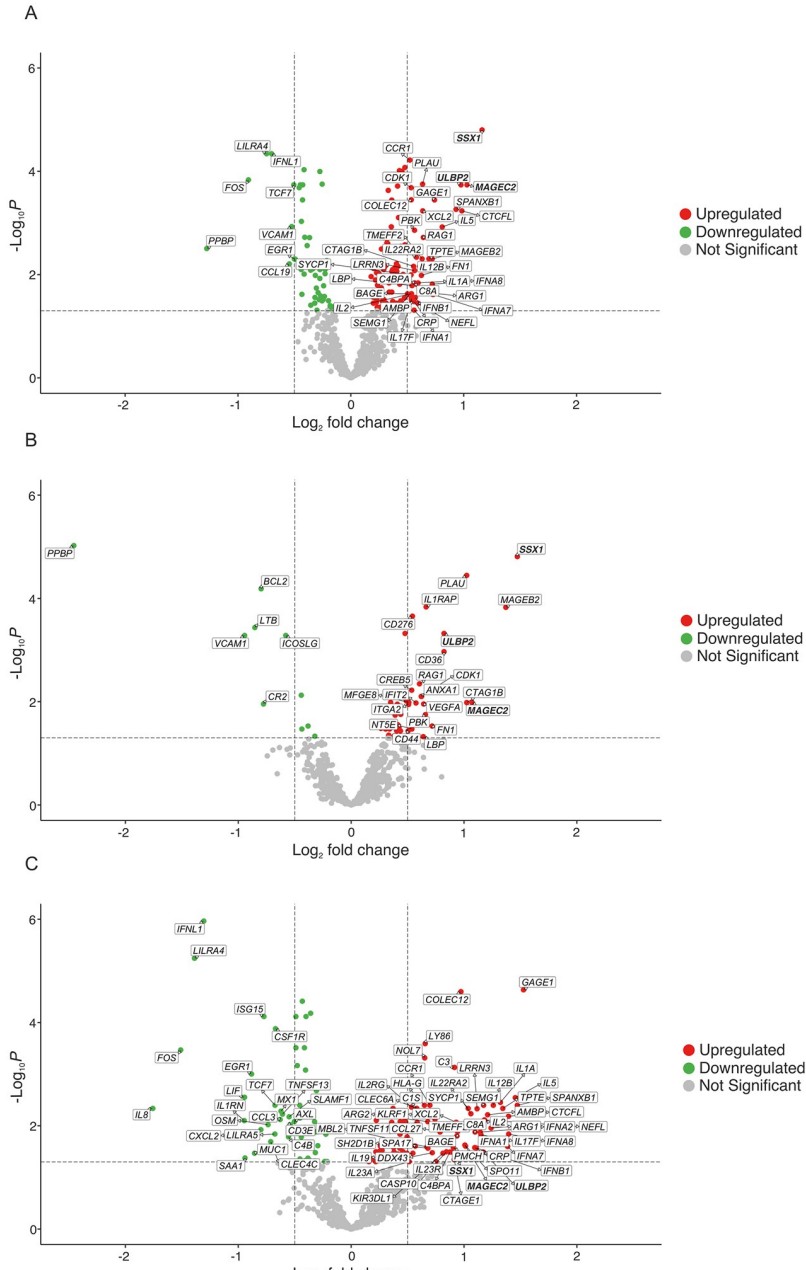

**Fig 4. Genes differentially expressed in short survival versus long survival patients, considered according to cancer type and cancer class.** Volcano plots showing significantly downregulated (the thresholds are FDR < 0.005; logFC < -0.5) and upregulated (the thresholds are FDR < 0.005; logFC > -0.5) genes in green and red, respectively. Here short survival patients are compared with long survival patients, so in all three panels genes found to be upregulated are genes that are upregulated in short survival patients, and genes found downregulated are genes that are downregulated in short survival patients. Grey indicates genes that are not significantly downregulated or upregulated. Log of fold change (LogFC) is on the x-axis and significance level (-log₁₀P) is on the y-axis. Panel A shows differentially expressed genes when all of the patients are considered (n = 515). Panel B shows differentially expressed genes when only patients with solid cancers are considered (n = 293), and panel C shows differentially expressed genes when only patients with blood cancers are considered (n = 222). In all panels, the significantly downregulated and upregulated genes are labelled with their Hugo Gene Nomenclature Committee (HGNC) gene symbols. The three genes (SSX1, MAGEC2 and ULBP2) that are found to be significantly differentially expressed in all three analyses are shown in bold.

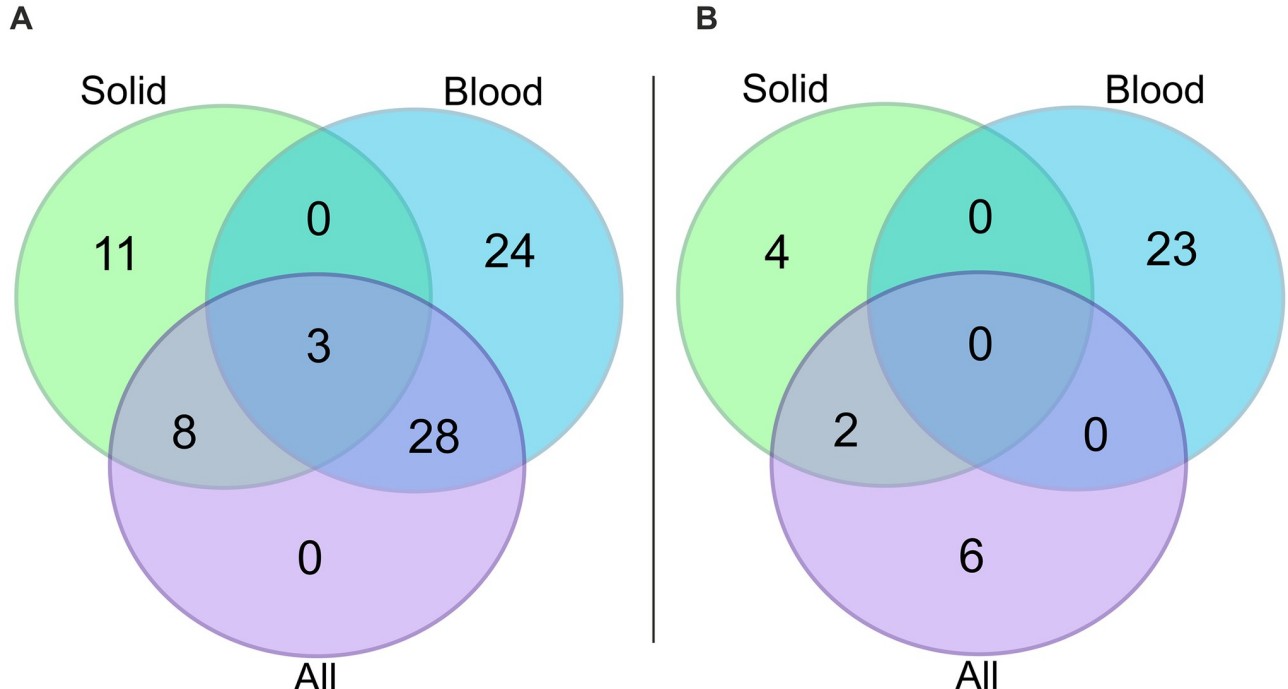

**Fig 5. Overlap of differentially expressed genes according to survival across cancer classifications.** Venn diagrams to visualize the numbers of significantly differentially expressed genes that overlap across the three differential gene expression analyses that were conducted: considering all patients (n = 515; mauve), considering only solid cancer patients (n = 293; green), and considering only blood cancer patients (n = 222; blue). Panel A shows the overlap of upregulated genes and panel B shows the overlap of downregulated genes across the three differential gene expression analyses.

of cancer-specific antigens including GAGE1, MAGEB2, MAGEC2 and SSX1; the other included cytokines (interferons and interleukins such as IFNA1, IFNB1, IL1, IL17, IL22) and ARG1, with a sub-network including CRP, FN1, PLAU and complement genes. Two protein-protein interaction networks were observed for genes upregulated in solid cancer (Fig 7): one network included products of cancer-specific genes such as MAGEB2, MAGEC2 and SSX1, as noted above for the whole cohort. The other network included products of genes associated with tumour growth and cell proliferation (FN1, VEGFA), tumour progression and metastasis (CD44, ITGA2) and immune inhibition (NT5E). Two protein-protein interaction networks relevant to immunotherapy were also observed for genes upregulated in blood cancers (Fig 8): one included the recurrent cancer antigens such as GAGE1, MAGEC2 and SSX1, whereas the other network mainly included products of interleukin-related genes involved in malignancy progression such as IL1, IL17, IL19, IL22 and IL23. Two additional networks, associated with complement activation (C1S, C3, C8A) and NK cell activation (KIR3DL1, KLRF1 and ULBP2), were observed. Expanding the protein-protein interaction analysis (S1–S3 Figs) revealed additional networks for the whole cohort: one network included ITGAV, LRP1, PLAT, and SERPINE1, all associated with tumour progression, cell invasion and metastasis; and the other included CDK1/2, NCAPG, PKB and CCN family genes, all associated with tumour progression and cell proliferation (S1 Fig). Similar observations were made for solid cancers (S2 Fig). Expanding the protein-protein interaction analysis for blood cancers revealed an additional network consisting of CD46, CFP, MASP1 and SERPINE1, correlated with short overall survival (S3 Fig).

## 3.6. Gene expression-based estimation of immune cell levels

Immune cell types and levels have long been linked to cancer outcomes [77]; high levels of specific immune cell types (e.g., CD8+ T cells, NK cells) within the tumour microenvironment

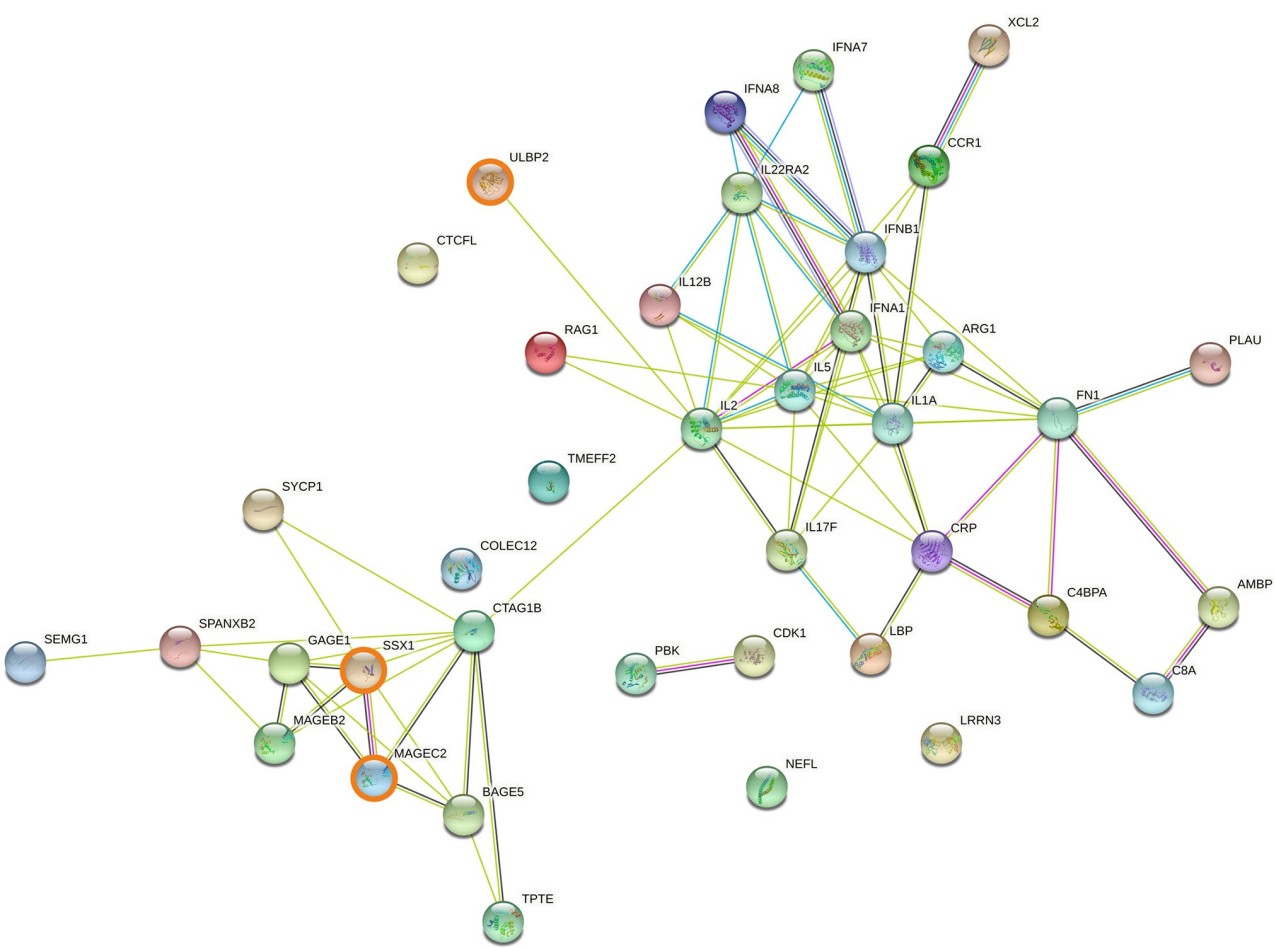

**Fig 6. Protein association network analysis using STRING in all cancer patients.** The predicted protein-protein interactions of significantly up-regulated genes (n = 39) in short survival cancer patients (FDR < 0.05, LogFC > 0.5) from the differential expression analysis incorporating both solid and blood cancer patients (n = 515). SSX1, MAGEC2 and ULBP2, the three genes found to be upregulated in both solid and blood patient cohorts, are shown in bold.

(TME) are generally linked to a higher probability of remaining cancer-free after surgery and longer overall survival [78, 79]. We used Cibersort [76] to estimate immune cell levels based on gene expression profiles. Across both solid and blood cancers, immune cell profiles included slightly higher levels of activated NK cells, CD4+ memory cells and CD8+ cells in long overall survival patients, and higher levels of M2 macrophages in short survival patients (Fig 9A; S6 Table). Long overall survival solid cancer patients showed substantially higher levels of CD4+ memory, resting mast cells, T regulatory (Treg) cells and activated NK cells (Fig 9B; S6 Table). Short overall survival solid cancer patients showed higher levels of M2 macrophages, monocytes and CD4+ naïve cells. Long overall survival blood cancer patients showed higher levels of Tregs, plasma cells, CD8+ and B naïve cells whereas short overall survival blood cancer patients showed higher levels of M2 macrophages, B memory cells, T follicular helper cells and γδ T cells (Fig 9C; S6 Table).

The different cancer types exhibited distinct immune cell profiles (Fig 9D). Activated NK cells were substantially more abundant in the cancer types associated with longer overall survival (breast, colon, lung and ovarian) whereas activated NK cells were decreased in the shorter overall survival cancer types (glioblastoma, melanoma, and head and neck cancers). Likewise,

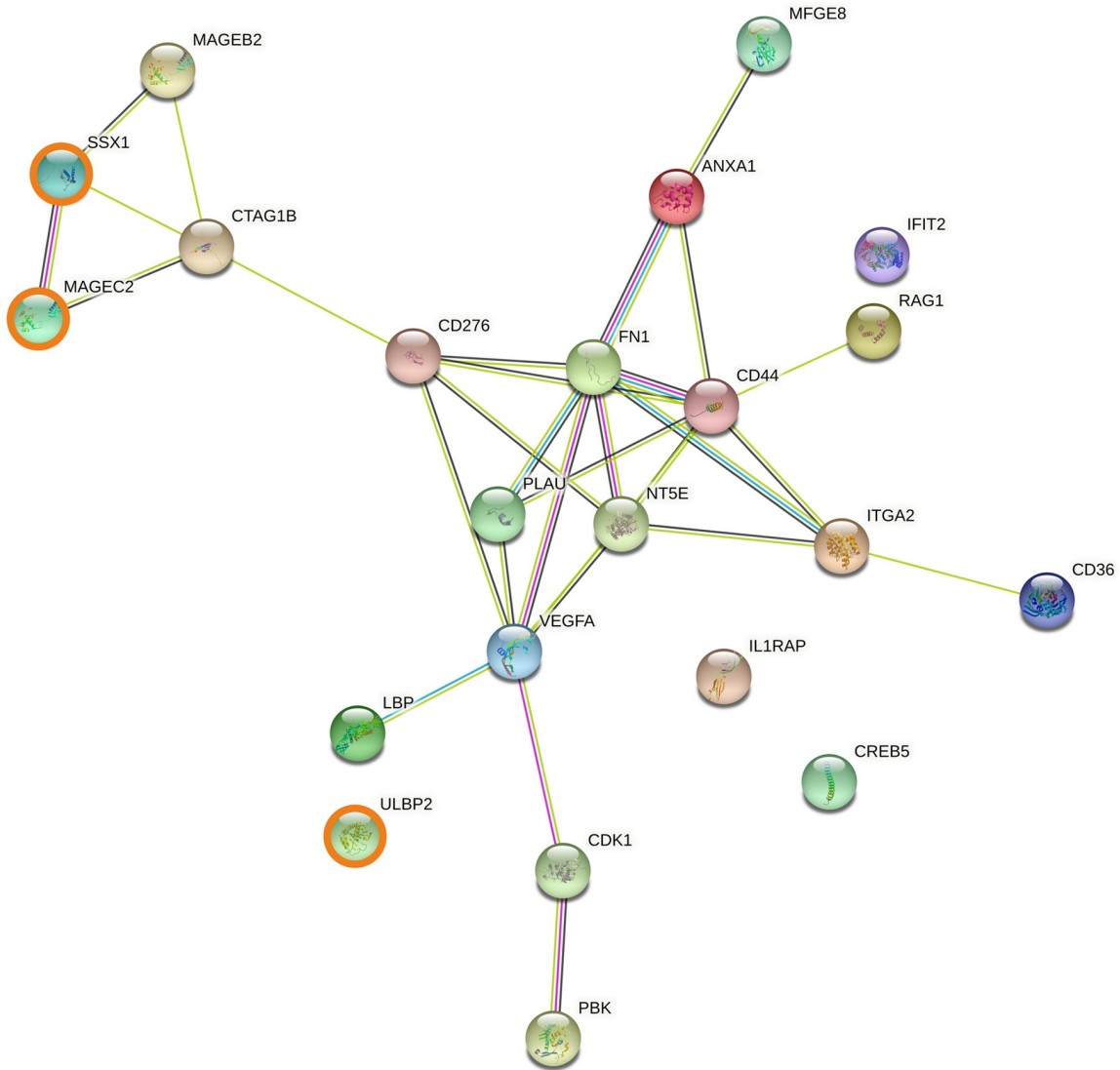

**Fig 7. Protein association network analysis using STRING in solid cancer patients.** The predicted protein-protein interactions of significantly up-regulated genes (FDR < 0.05, LogFC > 0.5) from the differential expression analysis incorporating only solid cancer patients (n = 293). SSX1, MAGEC2 and ULBP2, the three genes found to be upregulated in both solid and blood patient cohorts, are shown in bold.

CD8+ cells were more abundant in longer overall survival solid cancer types (breast, colon, lung and pancreatic) and markedly lower in short overall survival cancer types (glioblastoma, melanoma, and head and neck cancers). In contrast, immune cells associated with tumour promotion such as M2 macrophages were at higher levels within short overall survival cancer types (glioblastoma and melanoma), and γδ T cell levels showed a similar trend. Treg cell levels were unexpectedly higher in cancer types associated with longer overall survival (pancreatic, ovarian and breast cancers) and lower in cancer types associated with shorter survival (glioblastoma and melanoma). We did not consider the Cibersort results for the immune cell profiles of B cell and Hodgkin lymphomas due to the blood origin of these malignancies. Nevertheless, the relatively high B cell levels for B cell lymphoma support the reliability of Cibersort (S7 Table).

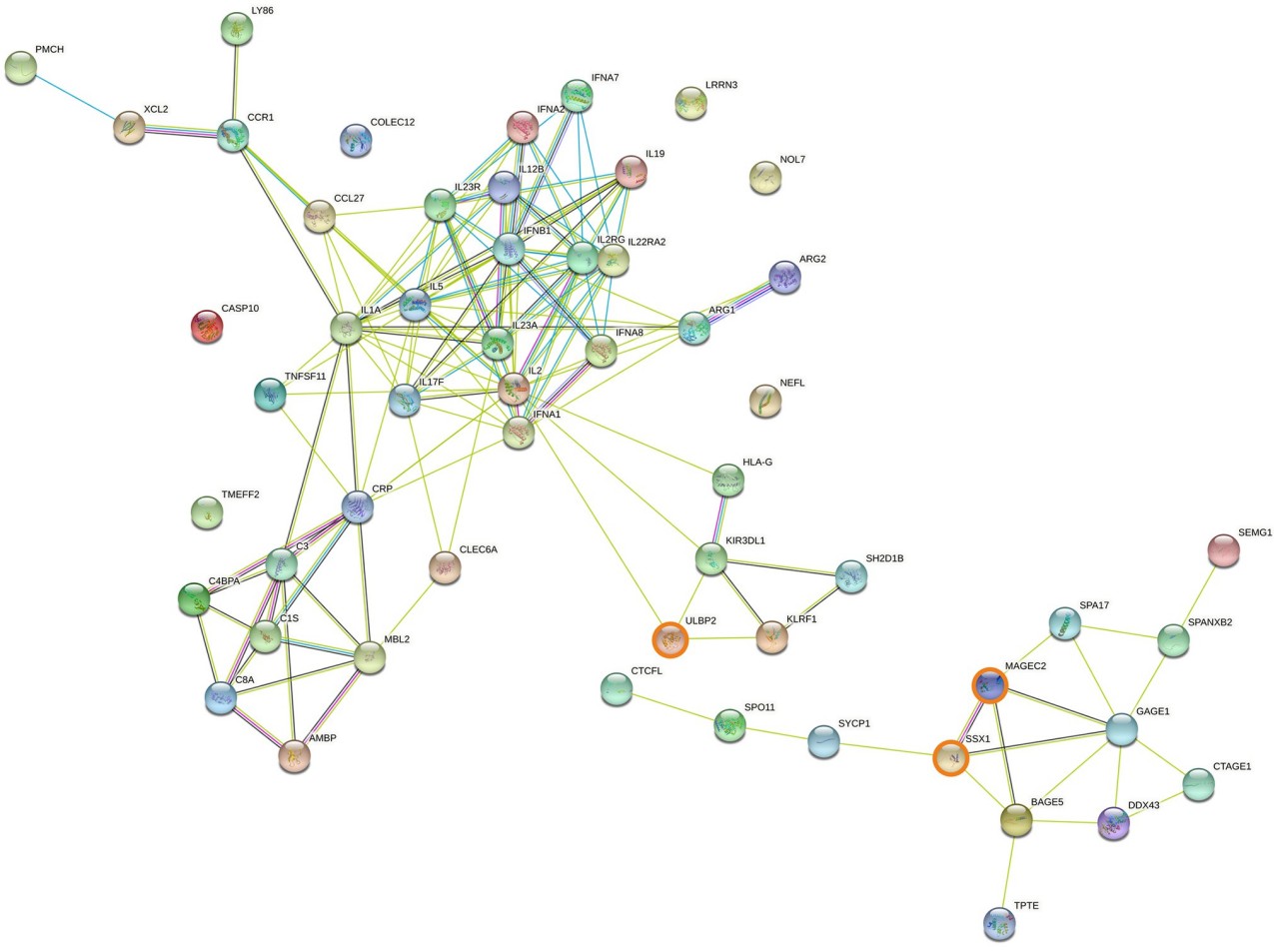

**Fig 8. Protein association network analysis using STRING in blood cancer patients.** The predicted protein-protein interactions of significantly up-regulated genes (FDR < 0.05, LogFC > 0.5) from the differential expression analysis incorporating only blood cancer patients (n = 222). SSX1, MAGEC2 and ULBP2, the three genes found to be upregulated in both solid and blood patient cohorts, are shown in bold.

## 4. Discussion

We have analysed immune-related gene expression data, generated with the NanoString Pan-Cancer Immune Profiling panel, for 515 cancer patients diagnosed with 10 different types of cancer. This NanoString panel enables direct characterisation of expression of 770 immune-related genes in solid and blood cancers, and indirect characterisation of immune cell infiltration in solid cancers. We consequently identified a prognostic immune gene signature shared by cancer patients with shorter overall survival, and, for solid cancer patients, we estimated immune infiltration profiles in relation to patient overall survival.

In terms of transcriptional signatures of overall survival, we identified a set of 39 genes upregulated in short overall survival cases across the cancer types examined. Based on protein-protein interaction analysis, this set of 39 genes includes two networks (Fig 6). One network comprises well known cancer antigens (e.g. GAGE1, MAGEB2, MAGEC2 and SSX1) associated with short overall survival and additional aggressive tumour properties [80–84]. The other network includes ARG1 which is expressed by immunosuppressive and tumourigenic M2 macrophages [85], and numerous cytokines. One subgroup of the cytokines is

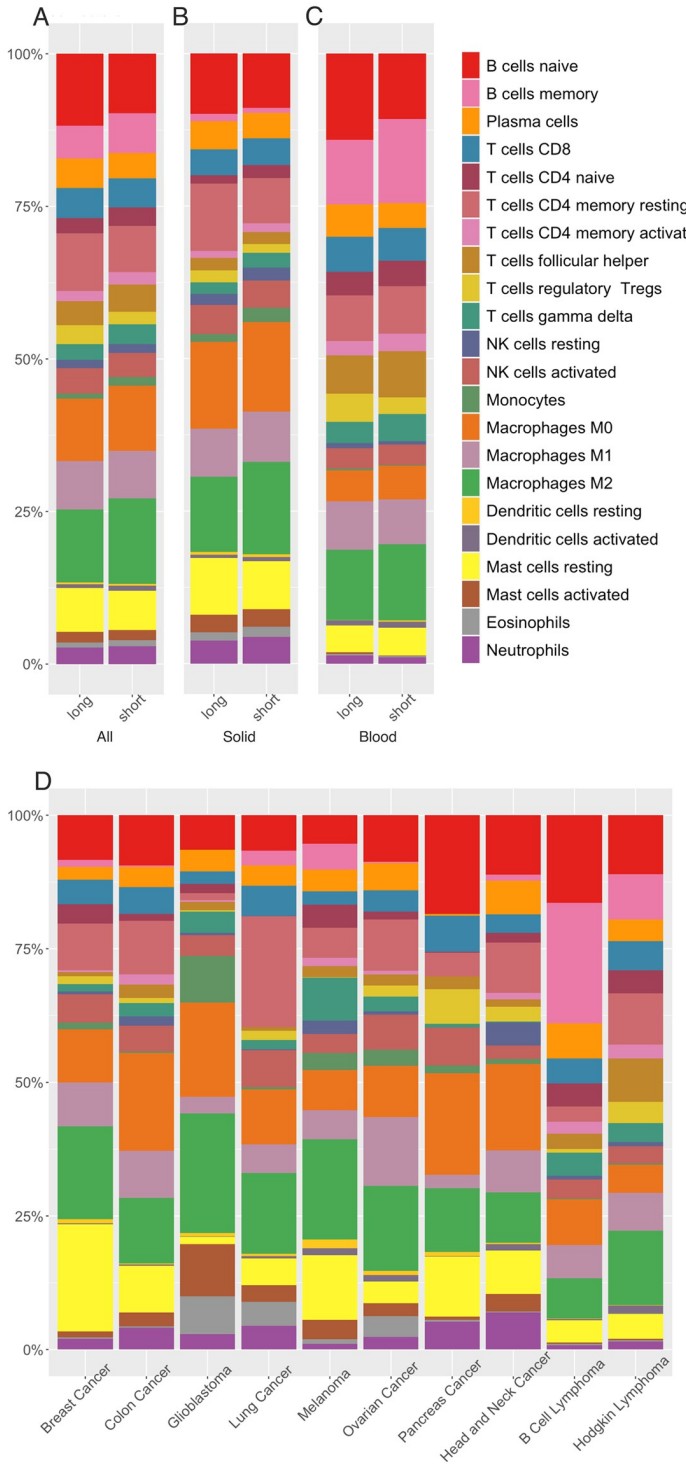

**Fig 9. Estimation of immune cell composition in 515 cancer patients using CIBERSORT.** Stacked bar plots in panels A, B and C show the CIBERSORT-derived immune profile of long versus short survival comparison of all, solid and blood cancer patients, respectively. Stacked bar plots in panel D show the immune profile of each cancer type under investigation. Different colours in bar plots represent different immune cell types according to the legend provided. The y-axis shows relative percentages of the cell types whereas the x-axis shows survival (long versus short) (panels A-C) or cancer types (panel D).

associated with tumour growth, inflammation, tumour progression and invasion (e.g. IL1, IL17, IL22) [85–92], and another subgroup includes genes associated with antitumour response (e.g. IL2, IL5, IL12 and IFN family) [93–98]. In addition, this protein-protein interaction network includes CRP, FN1 and PLAU, implicated in different ways with cancer risk or progression.

We observed marked differences in gene expression patterns between solid and blood malignancies (Figs 2B, 3, 4B, 4C and 5) so the association between differential gene expression and overall survival was analysed separately for solid and blood cancer patients. We identified 22 and 55 significantly upregulated genes in short overall survival solid and blood cancer patients, respectively (Fig 4). The protein-protein interaction networks (Figs 6–8) involving these two sets of upregulated genes are partly consistent with the patterns observed in the cross-cohort 39 gene set mentioned above. The protein-protein interaction network involving cancer antigens appears throughout the whole cohort, including solid and blood cancer patient groups, with some variations. In solid cancer patients, the cancer antigen interaction network includes CTAG1B, MAGEB2, MAGEC2 and SSX1; another network includes cell surface proteins such as CD44, CD276, ITGA2 and NT5E, extracellular matrix protein FN1, PLAU (uro-kinase) and growth factor VEGFA, associated with short overall survival, immune inhibition, metastasis, cell proliferation and tumour growth [99–105]. In blood cancer patients, the cancer antigen protein-protein interaction network includes a slightly broader range of genes such as BAGE5, CTAGE1, GAGE1, MAGEC2 and SSX1, and a cytokine-dominated network includes numerous interferons (IFNA1-7-8, IFNB1) and numerous interleukins associated with tumour growth, progression and metastasis (IL1A, IL17F, IL19, IL22, IL23) [85, 86, 88–92], plus ARG1 and ARG2. A third protein-protein interaction network in the blood cancer patient group includes genes associated with the complement system such as C1S, C3 and C8A, activation of which is mainly associated with pro-tumour effects [106].

Notably, upregulated genes in short survival cases across solid and blood cancer cohorts include a common immune gene signature comprising MAGEC2, SSX1 and ULBP2. These three genes are well known cancer antigens associated with poor prognosis, reduced free overall survival, and aggressive tumour behaviour; they are under investigation for immunotherapies in the form of cancer vaccines and CAR-T [80, 107–111]. In prostate cancer, for example, it has been observed that SSX1 expression is restricted to metastatic lesions in prostate cancer, with increased SSX1 expression observed in patients with advanced disease compared to healthy donors and patients with early-stage disease [81]. Additionally, SSX1 has been observed to be prognostic for both OS and progression-free survival (PFS) in patients affected by relapsed myeloma [80]. More recently, Qi et al reported increased invasiveness and conserved stem-like features in sarcoma cells with increased SSX1 expression [112]. MAGEC2 expression has been observed in advanced stages of different tumours, including myeloma and hepatocellular carcinoma (HCC), and is usually associated with poor prognosis [84]. MAGEC2-positive NSLCC patients, for example, showed a significant decrease in survival compared to MAGEC2-negative patients [84]. In HCC patients, MAGEC2 correlated with increased tumourigenesis and was associated with poor prognosis [83], with similar findings reported in breast cancer patients and melanoma patients [82, 113]. ULBP2 expression increased in serum and correlated with tumour progression in pancreatic cancer patients relative to healthy individuals [114], with similar observations in lung cancer patients [115]. In ovarian cancer patients, moreover, ULBP2 expression was correlated with poor prognosis [116].

Overall, our analysis shows that it is possible to capture a pan-cancer immune gene signature which could be particularly helpful for patients with short overall survival who might not benefit from standard therapy due to cancer aggressiveness. Given the numbers of immune-

related genes (n = 770) and patients (n = 515) included, and the variety of cancer types (n = 10), we believe the identified immune gene signature to be robust.

To further probe the relationships between immune-related genes and tumour microenvironment, we used Cibersort to estimate the proportions of immune cells across the whole cohort of patients, in solid and blood cancer types, and individual cancers. Patients with shorter overall survival exhibited a higher proportion of M2 macrophages, an immune cell population that exerts a protumorigenic effect and is commonly associated with short overall survival [117, 118]. γδ T cell levels were increased in patients with shorter overall survival and cancer types associated with short overall survival, corroborating the concept that γδ T cells are immunosuppressive and cancer-promoting [119, 120]. Tregs were unexpectedly more abundant in those patients and cancer types associated with longer overall survival (pancreatic, ovarian and breast cancer), and less abundant in those cancer types associated with shorter survival (glioblastoma and melanoma), underscoring the context-dependent role of Tregs in cancer [121]. Indeed, although Tregs have typically been linked with worse prognosis, recent evidence links Treg infiltration in the TME with improved clinical outcome in certain tumour types, underscoring the need to revisit the role of Treg cells in the TME [122]. CD8+ T cells, CD45RO+ (CD4+ T memory cells) and NK cells were slightly more abundant in patients and cancer types with longer overall survival. Histological investigations of human tumours have demonstrated the clinical importance of tumour infiltrating lymphocytes such as NK cells, revealing diverse levels of immune infiltration across patients and correlation of greater infiltration with improved clinical outcomes [123, 124]. The association between patient overall survival and robust immune infiltration has been well documented in ovarian, colorectal, breast, lung, oesophagal, melanoma, head and neck, urothelial and gallbladder cancers [77, 125, 126]. Consistent with our findings, these studies reported that a high density of CD8 + cytotoxic T lymphocytes and CD45RO+ memory T cells, plus CD3+ T cells, correlates with longer overall survival and disease-free overall survival. More recently, T cell density has increasingly been assessed together with T cell location–tumour centre or invasive margin—to provide a more detailed immune context [77].

Translating our observations towards clinical benefit will require validation of the identified prognostic immune gene signature of patient overall survival, for example through additional studies incorporating many more patients and an even wider spectrum of malignancies. Such extensive gene expression and immune cell infiltration profiling to monitor the activity of specific immune pathways and establish checkpoint inhibitor status could permit prediction of patient populations likely to benefit most from a particular immunotherapy. The NanoString PanCancer Immune Profiling panel, encompassing 770 immune-related genes and approximately 20 different biological processes, is a powerful tool in this quest to match tumour biology with therapeutic mode of action.

Our conclusions for blood cancers are more tentative than for solid cancers as only two types of lymphoma (large B cell and Hodgkin lymphoma) were included, although it is worth noting the relatively high Hodgkin and large B cell lymphoma patient numbers, 172 and 50 respectively, in our study. Hence our results might not be representative of transcriptional profiles in other lymphomas, leukaemias and myelodysplastic syndromes. Among other limitations, tumour stage was not available for all patients and therefore not included as a parameter in our analyses. Also, samples were collected and processed in different laboratories and thus potentially subjected to varying sample collection and RNA extraction protocols. Although not documented, some patients might have undergone neoadjuvant therapy before sample collection, thus altering the expression of immune-related genes. Lastly, it should be kept in mind that the reported immune cell levels are bioinformatics-based and lack experimental validation.

## 5. Conclusions

Alongside the recognition that every cancer type has unique features, and as oncology moves towards personalised treatments, there is increasing understanding of the prognostic and predictive strength of specific immune components [77, 127]. Patient transcriptomic profiling is increasingly recognized as an important tool for prediction of overall survival, for identification of potentially actionable transcriptomic targets or the need for treatment change, and for informing treatment customisation [128]. We consequently advocate investigation of immune determinants, either individually or in combination, across a large cohort of cancer patients as a strategy towards more accurate matching of patients with immunotherapeutic approaches; one outcome could be a set or sets of immune biomarkers that simultaneously reflect multiple tumours, tumour microenvironment and immune system features. This study, analysing immune transcriptional profiles across multiple malignancies and a range of survival characteristics, represents a step towards such a personalised approach.

## Supporting information

**S1 Fig. Expanded network analysis using STRING in short survival patients.** The panel shows expanded protein-protein interaction of up-regulated genes (n = 39) in short survival cancer patients (FDR < 0.05, LogFC > 0.5) within the whole cohort of cancer patients (n = 515). In bold those genes (SSX1, MAGEC2 and ULBP2) found shared across solid and blood patients.
(TIF)

**S2 Fig. Expanded network analysis using STRING in solid cancer patients.** The panel shows the expanded network for genes found significantly up-regulated (Log FC greater and equal to 0.5) for solid cancer patients. In bold those genes (SSX1, MAGEC2 and ULBP2) found shared across solid and blood patients.
(TIF)

**S3 Fig. Expanded network analysis using STRING in blood cancer patients.** The panel reports the expanded network for genes found significantly up-regulated (Log FC greater and equal to 0.5) for blood patients. In bold those genes (SSX1, MAGEC2 and ULBP2) found shared across solid and blood patients.
(TIF)

**S1 Table. Patients statistics.**
(DOCX)

**S2 Table. Cohort description and gene expression dataset numbers.**
(DOCX)

**S3 Table. List of genes found upregulated in short survival patients among the whole cohort of patients.**
(DOCX)

**S4 Table. List of genes found upregulated in short survival patients among solid patients.**
(DOCX)

**S5 Table. List of genes found upregulated in short survival patients among blood patients.**
(DOCX)

**S6 Table. CIBERSORT tables reporting values of 22 types of immune cell proportions for all patients, solid and blood cancer patients respectively in a long versus short survival**

**comparison.**
(DOCX)

**S7 Table. CIBERSORT table reporting values of 22 types of immune cell proportions according to cancer type.**
(DOCX)

## Acknowledgments

We thank Oxford Nanopore Technologies and the University of Bath for co-funding ADA's PhD studentship.

## Author Contributions

**Conceptualization:** Alberto D'Angelo.

**Data curation:** Navid Sobhani, Mattia Cinelli.

**Formal analysis:** Huseyin Kilili, Mattia Cinelli.

**Investigation:** Robert Chapman.

**Resources:** Ingeborg Tinhofer, Stefano Luminari, Benedetta Donati, Alessia Ciarrocchi, Riccardo Giannini, Roberto Moretto, Chiara Cremolini, Filippo Pietrantonio, Debora Bonazza, Robert Prins, Seung Geun Song, Yoon Kyung Jeon.

**Software:** Huseyin Kilili.

**Supervision:** Daniele Generali, Benedetta Donati, Giuseppina Pisignano, Araxi O. Urrutia.

**Writing – original draft:** Alberto D'Angelo.

**Writing – review & editing:** Stefan Bagby, Araxi O. Urrutia.

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
