## [Decision Letter · Decision Letter 0]

28 Sep 2022

PONE-D-22-19319Immune-related pan-cancer gene expression signatures of patient survival revealed by NanoString-based analysesPLOS ONE

Dear Dr. D'Angelo,

Thank you for submitting your manuscript to PLOS ONE. After careful consideration, we feel that it has merit but does not fully meet PLOS ONE’s publication criteria as it currently stands. Therefore, we invite you to submit a revised version of the manuscript that addresses the points raised below by the reviewers during the review process.

We look forward to receiving your revised manuscript.

Kind regards,

Surinder K. Batra

Academic Editor

PLOS ONE

Journal Requirements:

Reviewers' comments:

Reviewer's Responses to Questions

**Comments to the Author**

1. Is the manuscript technically sound, and do the data support the conclusions?

Reviewer #1: Yes

Reviewer #2: Yes

2. Has the statistical analysis been performed appropriately and rigorously? 

Reviewer #1: Yes

Reviewer #2: Yes

3. Have the authors made all data underlying the findings in their manuscript fully available?

Reviewer #1: Yes

Reviewer #2: Yes

4. Is the manuscript presented in an intelligible fashion and written in standard English?

Reviewer #1: Yes

Reviewer #2: Yes

5. Review Comments to the Author

Reviewer #1: The research article titled “Immune-related pan-cancer gene expression signatures of patient survival revealed by NanoString-based analyses.”, by D’Angelo et al, describes about the gene expression anlysis in multiple cancers to identify the novel immune related gene signature tht could be potentially used in cancer prognosis and therapy response analysis. Overall, the manuscript is nicely written, and methods used to generate the data is robust and machine based. Authors have highlighted the novel gene signature associated with solid tumors of patients with shorter survival. Results and discussion are appropriately presented in the manusctipt. Following are the minor comments for authors that need be addressed before considering the manuscript for the publication in the journal PLOS One. Authors are suggested to revise the manuscript to address the following comments:

1. Line 53-56: Please change the sentence as “A better understanding of transcriptional changes in immune cell related genes associated with cancer progression, and their significance in disease prognosis is therefore needed.”

2. Line 62-63: Change the sentence as “Most of the genes identified have previously been reported as relevant in one or more cancer types.”

3. Line 79-82: Sentence is complex and can be simplified as “Such variable success is probably linked to the complexity of the tumor microenvironment, which involves cell-cell interactions among multiple cell types, with accompanying dynamic genomic and epigenetic characteristics (7–9).

4. Section 3.2: Authors are suggested to explain the significance of clustering in the result section. It will help in understanding the data more clearly.

5. Overall, figure legends are too short and not found sufficient to explain the figures. For example, in figure legend 3, it is desired from authors to elaborate the figure legend, mentioning the statistical analysis in the figure a, and mention the details for the color scale of heatmap. In total, all figure legends need more description for better understanding.

6. As GAGE1, MAGEC2 and SSX1, is novel finding as a cluster for solid malignancies, authors are suggested to include more description for these genes and how these can be suitable for disease prognosis.

7. Figure 6-8: legends need to be more descriptive of figure and require a rewrite as they look like description of result and methods, rather than a description of figure.

8. Line 321-327: Please describe what does it means to have high or low immune cell types in solid tumors, in context of survival and treatment response.

9. Line 328-340: Immune cell analysis of long and short surviving patients have been compared by the authors, comparing different cancers. However, in a single cancer type, long and short survival patients have not been compared in a single cancer type. Authors are suggested to include that analysis to validate the gene signature, reported by the authors.

Reviewer #2: The manuscript “Immune-related pan-cancer gene expression signature of patients survival revealed by nanostring-based analysis” submitted by D’Angelo et al., investigated the modulations in the immune signature in multiple solid and hematological malignancies using available datasets generated using NanoString Pancancer gene panel consisting of 770 immune cells-related genes. The analysis demonstrated that 39 genes were upregulated in the patients with overall shorter survival. Three genes panel, MAGEC2, SSX1, and ULBP2, were upregulated both in solid and hematological malignancies. Further using the Cibersort analysis, the authors demonstrated that tumors derived from the patients with shorter survival showed a higher proportion of M2 macrophages and γδ T cells, whereas patients with longer overall survival showed a higher proportion of CD8+ T cells, CD4+ T memory cells, NK cells, and T regulatory cells. Overall, the manuscript is well written. I have the following minor concern that needs to be addressed before the acceptance of the manuscript.

Minor Concerns

1. It is important to include information on the stages of the different tumor types included in the study if available. This is important because tumors like pancreatic cancer are generally resected at the early stages, as most patients are presented with metastatic disease and are not eligible for resection.

2. Include a brief discussion on MAGEC2, SSX1, and ULBP2 and the reason for their association with shorter survival. Similarly, it is important to highlight the cancers where the presence of T-regulatory cells associate with longer survival, and if you separate those cancers, are T-reg cells associated with shorter survival?

6. PLOS authors have the option to publish the peer review history of their article (what does this mean?). If published, this will include your full peer review and any attached files.

Reviewer #1: **Yes: **Shailendra Gautam

Reviewer #2: No

---

## [Author Response · Author response to Decision Letter 0]

29 Oct 2022

Thank you for raising this point. The manuscript should now be formatted accordingly. 

Thank you for raising this point. We have modified the data availability statement accordingly. Kindly note that we did not generate any data ourselves and part of the data used for the analysis was downloaded from publicly available online repositories. When a dataset was not publicly available, we emailed the contact person and asked them to share their data. A list of GSE numbers is provided as a table in Supplementary Material as specified in the updated version of our Data Availability statement.

Thank you for your suggestion. We have carefully checked our reference list and we believe there to be no substantial issues with the references listed within the manuscript.

Reviewers' comments:

Reviewer's Responses to Questions 

Comments to the Author

1. Is the manuscript technically sound, and do the data support the conclusions?

Reviewer #1: Yes

Reviewer #2: Yes

2. Has the statistical analysis been performed appropriately and rigorously? 

Reviewer #1: Yes

Reviewer #2: Yes

3. Have the authors made all data underlying the findings in their manuscript fully available?

Reviewer #1: Yes

Reviewer #2: Yes

4. Is the manuscript presented in an intelligible fashion and written in standard English?

Reviewer #1: Yes

Reviewer #2: Yes

5. Review Comments to the Author

Reviewer #1: The research article titled “Immune-related pan-cancer gene expression signatures of patient survival revealed by NanoString-based analyses.”, by D’Angelo et al, describes about the gene expression anlysis in multiple cancers to identify the novel immune related gene signature tht could be potentially used in cancer prognosis and therapy response analysis. Overall, the manuscript is nicely written, and methods used to generate the data is robust and machine based. Authors have highlighted the novel gene signature associated with solid tumors of patients with shorter survival. Results and discussion are appropriately presented in the manusctipt. Following are the minor comments for authors that need be addressed before considering the manuscript for the publication in the journal PLOS One. Authors are suggested to revise the manuscript to address the following comments:

1. Line 53-56: Please change the sentence as “A better understanding of transcriptional changes in immune cell related genes associated with cancer progression, and their significance in disease prognosis is therefore needed.”

Thank you for your suggestion. We have modified the sentence accordingly.

2. Line 62-63: Change the sentence as “Most of the genes identified have previously been reported as relevant in one or more cancer types.”

Thank you for your suggestion. We have modified the sentence accordingly.

3. Line 79-82: Sentence is complex and can be simplified as “Such variable success is probably linked to the complexity of the tumor microenvironment, which involves cell-cell interactions among multiple cell types, with accompanying dynamic genomic and epigenetic characteristics (7–9).

Thank you for your suggestion. We have modified the sentence accordingly.

4. Section 3.2: Authors are suggested to explain the significance of clustering in the result section. It will help in understanding the data more clearly.

Thank you for the suggestion. We have addressed this in the text.

5. Overall, figure legends are too short and not found sufficient to explain the figures. For example, in figure legend 3, it is desired from authors to elaborate the figure legend, mentioning the statistical analysis in the figure a, and mention the details for the color scale of heatmap. In total, all figure legends need more description for better understanding.

Thank you for raising this point. We have expanded all figure captions/legends.

6. As GAGE1, MAGEC2 and SSX1, is novel finding as a cluster for solid malignancies, authors are suggested to include more description for these genes and how these can be suitable for disease prognosis.

Thank you for the suggestion. We have commented on the prognostic role of these genes in the “Discussion” section.

7. Figure 6-8: legends need to be more descriptive of figure and require a rewrite as they look like description of result and methods, rather than a description of figure.

Thank you for the suggestion. We have addressed this in the text.

8. Line 321-327: Please describe what does it means to have high or low immune cell types in solid tumors, in context of survival and treatment response.

Thank you for the suggestion. We have included a short comment about how immune cell type levels can influence the survival of cancer patients (lines 367-368).

9. Line 328-340: Immune cell analysis of long and short surviving patients have been compared by the authors, comparing different cancers. However, in a single cancer type, long and short survival patients have not been compared in a single cancer type. Authors are suggested to include that analysis to validate the gene signature, reported by the authors.

Thank you for raising this point. In terms of a long versus short survival comparison within each cancer type, we originally ran these analyses but did not achieve statistical validity due to the discrepancy in the number of patients for each cancer type. Indeed, the number of patients was very limited for pancreatic cancer (n=7) and lung cancer (n=17), and much higher for colon cancer (n=89) and Hodgkin lymphoma (n=172). For this reason, we aimed to investigate differences in survival across all patients and also across patients split into two groups - solid and blood malignancy types.

 

Reviewer #2: The manuscript “Immune-related pan-cancer gene expression signature of patients survival revealed by nanostring-based analysis” submitted by D’Angelo et al., investigated the modulations in the immune signature in multiple solid and hematological malignancies using available datasets generated using NanoString Pancancer gene panel consisting of 770 immune cells-related genes. The analysis demonstrated that 39 genes were upregulated in the patients with overall shorter survival. Three genes panel, MAGEC2, SSX1, and ULBP2, were upregulated both in solid and hematological malignancies. Further using the Cibersort analysis, the authors demonstrated that tumors derived from the patients with shorter survival showed a higher proportion of M2 macrophages and γδ T cells, whereas patients with longer overall survival showed a higher proportion of CD8+ T cells, CD4+ T memory cells, NK cells, and T regulatory cells. Overall, the manuscript is well written. I have the following minor concern that needs to be addressed before the acceptance of the manuscript.

Minor Concerns

1. It is important to include information on the stages of the different tumor types included in the study if available. This is important because tumors like pancreatic cancer are generally resected at the early stages, as most patients are presented with metastatic disease and are not eligible for resection.

Thank you for raising this point. Unfortunately, tumour stage information was not available for all the patients under evaluation. We have now mentioned this limitation in our “Discussion” section (line 505). However, we are in the process of re-running our analyses for a future manuscript with an increased number of patients and also only for those patients with information such as age, gender and tumour stage available.

2. Include a brief discussion on MAGEC2, SSX1, and ULBP2 and the reason for their association with shorter survival. Similarly, it is important to highlight the cancers where the presence of T-regulatory cells associate with longer survival, and if you separate those cancers, are T-reg cells associated with shorter survival?

Thank you for your suggestion. We have expanded the discussion for genes SSX1, MAGEC2 and ULBP2, highlighting their correlation with poor survival.

With regard to the Treg population, we have specified the cancer types in which the Treg population was unexpectedly observed to be increased and decreased (lines 388-389). In lines 476-479, within the Discussion section, we have specified more accurately that our results concerning the Treg cell population contrast with a large part of the literature. 

In terms of Tregs presence for a long versus short survival comparison within each cancer type, we originally ran these analyses but did not achieve statistical validity due to the discrepancy of the number of patients for each cancer type; the number of patients was limited for pancreatic cancer (n=7) and lung cancer (n=17), and much higher for colon cancer (n=89) and Hodgkin lymphoma (n=172). 

6. PLOS authors have the option to publish the peer review history of their article (what does this mean?). If published, this will include your full peer review and any attached files.

Do you want your identity to be public for this peer review? For information about this choice, including consent withdrawal, please see our Privacy Policy.

Reviewer #1: Yes: Shailendra Gautam

Reviewer #2: No

---

## [Decision Letter · Decision Letter 1]

28 Dec 2022

Immune-related pan-cancer gene expression signatures of patient survival revealed by NanoString-based analyses

PONE-D-22-19319R1

Dear Dr. D'Angelo,

We’re pleased to inform you that your manuscript has been judged scientifically suitable for publication and will be formally accepted for publication once it meets all outstanding technical requirements.

Kind regards,

Surinder K. Batra

Academic Editor

PLOS ONE

Additional Editor Comments (optional):

Reviewers' comments:

Reviewer's Responses to Questions

**Comments to the Author**

1. If the authors have adequately addressed your comments raised in a previous round of review and you feel that this manuscript is now acceptable for publication, you may indicate that here to bypass the “Comments to the Author” section, enter your conflict of interest statement in the “Confidential to Editor” section, and submit your "Accept" recommendation.

Reviewer #1: All comments have been addressed

2. Is the manuscript technically sound, and do the data support the conclusions?

Reviewer #1: Yes

3. Has the statistical analysis been performed appropriately and rigorously? 

Reviewer #1: Yes

4. Have the authors made all data underlying the findings in their manuscript fully available?

Reviewer #1: Yes

5. Is the manuscript presented in an intelligible fashion and written in standard English?

Reviewer #1: Yes

6. Review Comments to the Author

Reviewer #1: Authors have revised the manuscript and all the comments have been addressed. The manuscript can be considered for publication.

7. PLOS authors have the option to publish the peer review history of their article (what does this mean?). If published, this will include your full peer review and any attached files.

Reviewer #1: **Yes: **Shailendra K Gautam

---

## [Editor Report · Acceptance letter]

6 Jan 2023

PONE-D-22-19319R1 

Immune-related pan-cancer gene expression signatures of patient survival revealed by NanoString-based analyses 

Dear Dr. D'Angelo:

I'm pleased to inform you that your manuscript has been deemed suitable for publication in PLOS ONE. Congratulations! Your manuscript is now with our production department. 

Kind regards, 

on behalf of

Prof. Surinder K. Batra 

Academic Editor

PLOS ONE